# Restless and Uncertain: Robust Policies for Restless Bandits via Deep Multi-Agent Reinforcement Learning

**Jackson A. Killian**[1]     **Lily Xu**[1]     **Arpita Biswas**[1,2]     **Milind Tambe**[1,2]

[1]Computer Science, Harvard University, Cambridge, MA, USA
[2]Center for Research on Computation and Society, Harvard University, Cambridge, MA, USA

## Abstract

We introduce robustness in *restless multi-armed bandits* (RMABs), a popular model for constrained resource allocation among independent stochastic processes (arms). Nearly all RMAB techniques assume stochastic dynamics are precisely known. However, in many real-world settings, dynamics are estimated with significant *uncertainty*, e.g., via historical data, which can lead to bad outcomes if ignored. To address this, we develop an algorithm to compute minimax regret–robust policies for RMABs. Our approach uses a double oracle framework (oracles for *agent* and *nature*), which is often used for single-process robust planning but requires significant new techniques to accommodate the combinatorial nature of RMABs. Specifically, we design a deep reinforcement learning (RL) algorithm, DDLPO, which tackles the combinatorial challenge by learning an auxiliary "λ-network" in tandem with policy networks per arm, greatly reducing sample complexity, with guarantees on convergence. DDLPO, of general interest, implements our reward-maximizing agent oracle. We then tackle the challenging regret-maximizing nature oracle, a non-stationary RL challenge, by formulating it as a multi-agent RL problem between a policy optimizer and adversarial nature. This formulation is of general interest—we solve it for RMABs by creating a multi-agent extension of DDLPO with a shared critic. We show our approaches work well in three experimental domains.

## 1 INTRODUCTION

Restless multi-armed bandits (RMABs), a model for constrained resource allocation among $N$ independent stochastic processes (arms), are widely studied. Traditionally a *binary-action* problem, in which a planner decides whether or not to act on each of $N$ arms, here we consider the *multi-action* generalization [Killian et al., 2021b, Glazebrook et al., 2011] which more accurately captures challenging real-world planning problems. Salient examples of RMABs include scheduling [Bagheri and Scaglione, 2015, Yang et al., 2018], machine replacement [Glazebrook et al., 2006, Ruiz-Hernández et al., 2020], aerial vehicle routing [Le Ny et al., 2008], anti-poaching patrol planning [Qian et al., 2016], and healthcare [Lee et al., 2019, Mate et al., 2020]. While these works have established important theoretical foundations, they share one key limitation: assuming stochastic dynamics are precisely known. Having exact knowledge of dynamics is impossible in many real-world problems. For example, in healthcare intervention planning, the probability that a patient will adhere to treatment after receiving an intervention is not perfectly known *a priori*; in anti-poaching patrol planning, the probability of finding a poacher's snare at some location is not known with certainty.

Accordingly, methods have been developed to learn RMAB policies *online*, assuming no *a priori* knowledge [Jung et al., 2019, Wang et al., 2020]. However, these methods require tens of thousands of samples to converge to good policies which is prohibitive for many real-world problems, e.g., in finite-length treatment settings such as tuberculosis [Mate et al., 2020] with only a few dozen rounds. Instead, real-world planners must make the most of noisy data at hand, estimating dynamics from historical data or consulting experts, inducing significant *uncertainty*. RMAB techniques can be used to plan with point estimates, but we show that ignoring uncertainty can lead to arbitrarily bad policies.

To address these shortcomings and push RMABs toward wider real-world applicability, we introduce *Robust RMABs*, a generalization of RMABs which allows stochastic dynamics to be specified as uncertainty intervals, rather than point estimates. This new problem is very computationally demanding, adding a combinatorial layer of complexity onto an already PSPACE-hard problem [Papadimitriou and Tsitsiklis, 1994]. Addressing this complexity gives rise to a

*Accepted for the 38th Conference on Uncertainty in Artificial Intelligence* (UAI 2022).

rich set of challenges that necessitates the design of new techniques that not only help solve the robust objective we analyze, but also are of general interest to RMAB research.

Concretely, we plan under a *minimax regret* objective, using a double oracle (DO) framework [McMahan et al., 2003] that has seen success in problems involving a *single* Markov decision process (MDP) [Xu et al., 2021]. The DO approach casts the robust planning problem as a zero-sum game between an *agent* oracle and adversarial *nature* oracle. However, existing techniques fail for any non-trivially sized RMABs since the state and action spaces grow combinatorially in the number of arms $N$ and resource constraint $B$, respectively. Specifically, given $S$-sized state spaces for each arm, the full combinatorial problem has state space of size $S^N$ and action space–and thus policy-network output– of size $\binom{N}{B}$ (for binary-action RMAB; action space is larger with multi-action). At this size, we found that directly applying Xu et al. [2021] to solve the full combinatorial problem as a single process fails to learn good policies for RMABs as small as $N = 5$ arms, with $B = 3$ and $S = 2$. Moreover, under the minimax regret objective, the nature oracle is a particularly difficult challenge as it requires jointly searching the RMAB policy space and the continuous, uncertain space of transition probabilities. Previously, this objective has been posed as a non-stationary RL problem and solved heuristically with a single policy network [Xu et al., 2021]. We improve the nature oracle by formulating it as a multi-agent RL problem and develop a novel solution method for RMABs. In summary, our contributions are:

1. We introduce the Robust RMAB problem with interval uncertainty over arm dynamics and develop techniques to solve a minimax regret objective via double oracle.

2. To enable the DO approach, we introduce DDLPO, a novel deep RL algorithm for RMABs, of general interest. DDLPO tackles the combinatorial complexity of RMABs by learning an auxiliary "$\lambda$-network" in tandem with individual arm policy networks, which greatly reduces training sample complexity. The procedure implements the reward-maximizing agent oracle, has convergence guarantees, and solves RMABs with multiple action types [Killian et al., 2021b, Glazebrook et al., 2011], the first deep RL procedure to do so. DDLPO also easily extends to more general weakly-coupled MDPs [Adelman and Mersereau, 2008, Hawkins, 2003] and enables computing continuous-action policies, a previously unstudied RMAB direction.

3. We formulate the non-stationary regret-maximizing nature oracle as a multi-agent RL (MARL) problem, a framework of potential general interest in robust planning. We solve this problem in the combinatorially hard RMAB setting by extending DDLPO to include a shared critic and a continuous-action policy network for nature's selection of the uncertain transition dynamics.

## 2 RELATED WORK

**RMABs** The reward-maximizing, binary-action RMAB problem was introduced by Whittle [1988]. His widely used Whittle index policy [Mate et al., 2020, Glazebrook et al., 2006, Bagheri and Scaglione, 2015] is asymptotically optimal under *indexability* [Weber and Weiss, 1990]. Glazebrook et al. [2011] and Hodge and Glazebrook [2015] extended the Whittle index to multi-action RMABs with special monotonic structure, while Killian et al. [2021b] gave a more general Lagrange-based method. Hawkins [2003] studied methods for weakly coupled Markov decision processes (WCMDP), which generalize multi-action RMABs to have multiple constraints, and propose Lagrangian solutions for small problems. Adelman and Mersereau [2008] and Gocgun and Ghate [2012] followed by providing better solutions to WCMDPs but sacrifice scalability. All these works assumed precise knowledge of stochastic dynamics. Some recent works have studied online RMABs with unknown dynamics but all have prohibitively large sample complexity [Gafni and Cohen, 2020, Jung and Tewari, 2019, Biswas et al., 2021, Killian et al., 2021a]. None consider robust planning under environment uncertainty, which we address.

Our work also relates to learning algorithms for *stochastic* multi-armed bandit (MAB) problems [Min et al., 2020, Boutilier et al., 2020, Kuleshov and Precup, 2000]. However, since stochastic MABs follow a stateless reward process, learning algorithms utilize the fact that the true optimal policy simply selects the top $B$ reward-producing arms each round. Conversely, the arms in restless MABs have reward processes that follow MDPs, so the top $B$ arms to play each round is state- and action-dependent and constantly evolving, making both the learning and the planning problems much more challenging, and which our algorithms address.

**RL for RMABs** A few recent works learn Whittle indices for indexable binary-action RMABs using (i) deep RL (DRL) [Nakhleh et al., 2021] and (ii) tabular Q-learning [Biswas et al., 2021, Fu et al., 2019, Avrachenkov and Borkar, 2022]. Killian et al. [2021a] take tabular Q-learning to the multi-action setting. In contrast, our DRL approach provides a more general solution to binary and multi-action RMAB domains, not requiring indexability or problem structure, and is far more scalable than tabular methods. We are also the first to handle continuous-action RMABs, key to the nature oracle. Also related is the space of combinatorial RL. However, most existing algorithms consider single-shot problems, e.g., traveling salesman [Kool et al., 2019, Khalil et al., 2017], which lack a notion of future state that is critical to solving any version of RMAB, and none accommodate the general cost/budget structure of multi-action RMAB [Song et al., 2019]; our methods address these limitations.

**Robust planning** Work on robust planning in RL mainly focuses on maximin reward via robust adversarial RL [Pinto

et al., 2017] or multi-agent RL (MARL) [Lanctot et al., 2017, Li et al., 2019], but maximin reward leads to overly conservative policies [Nguyen et al., 2014]. The minimax regret criterion [Braziunas and Boutilier, 2007] avoids this pitfall, but this objective is challenging with very large or continuous strategy spaces. This can be addressed with the DO approach proposed by McMahan et al. [2003] which explores a small subset of strategies while still guaranteeing optimal convergence [Gilbert and Spanjaard, 2017]. Subsequently, DO has been extended to optimize MARL problems with multiple selfish agents [Lanctot et al., 2017]. Recently, Xu et al. [2021] used DO to solve a single Markov decision process (MDP) minimax-regret planning problem and used RL to implement the oracles. However, when applied to RMABs, the number of outputs in their policy network grows exponentially, as does the size of the state space being learned, both of which require prohibitively long training times beyond trivially sized RMABs. Accordingly, we found that their RL algorithms failed to scale past $N = 5$ arms and $S = 2$ states, whereas we show in Sec. 5 that our algorithms solve problems that are orders of magnitude larger. Additionally, their approach is designed only for continuous state/action spaces, whereas our approach can find robust policies for any combination of discrete *or* continuous state/action spaces. We accomplish this via our novel formulation of the nature oracle as a MARL problem, which decomposes the causes of non-stationarity, i.e., agent and nature, and learn them with separate networks.

## 3   PRELIMINARIES

We consider the multi-action RMAB setting with $N$ arms [Killian et al., 2021b, Glazebrook et al., 2011], which generalizes classical binary-action RMABs [Whittle, 1988].[1] Each arm $n \in [N]$ follows an MDP $(\mathcal{S}_n, \mathcal{A}_n, \mathcal{C}_n, T_n, R_n, \beta)$, where $\mathcal{S}_n$ is a set of finite, discrete states; $\mathcal{A}_n$ is a set of finite, discrete actions; $\mathcal{C}_n : \mathcal{A}_n \to \mathbb{R}$ defines action costs, where $\mathcal{C}_n[0] = 0$ encodes a no-cost "passive action" for all arms; $T_n : \mathcal{S}_n \times \mathcal{A}_n \times \mathcal{S}_n \to [0, 1]$ gives the probability of transitioning from one state to another given an action; $R_n : \mathcal{S}_n \to \mathbb{R}$ is a reward function; and $\beta \in [0, 1)$ is the discount factor. For ease of exposition, let $\mathcal{S}_n, \mathcal{A}_n, \mathcal{C}_n,$ and $R_n$ be the same for all $n \in [N]$, and thus drop the subscript $n$, though all methods apply to the general case. Let $s$ be an $N$-length vector of states over all arms and let $\boldsymbol{A} \in \{0,1\}^{N \times |\mathcal{A}|}$ be a decision matrix that one-hot-encodes the action taken on each arm. The planner computes policies $\pi$ which map states $s$ to actions $\boldsymbol{A}$ with the constraint that the sum cost of actions is less than a budget $B$ in every round $t \in [H]$.

We extend multi-action RMABs to the robust setting in

which the exact transition probabilities are unknown. Instead, the transition dynamics $T_n$ of each arm $n \in [N]$ are determined by a set of parameters $\omega_n \in \Omega_n$, each within a given interval uncertainty $\overline{\underline{\omega}}_n := [\underline{\omega}_n, \overline{\omega}_n]$. Let $\omega$ be a given parameter setting such that $\omega_n \in \overline{\underline{\omega}}_n$ for all $n \in [N]$. Let $G(\pi, \omega) = \mathbb{E}[\sum_{t=1}^{H} \beta^t \sum_{n \in [N]} R(\boldsymbol{s}_t^n) \mid \pi, \omega]$ be the planner's expected discounted reward under $\pi$ and $\omega$, where $\boldsymbol{s}_n^t$ is the state of arm $n$ at time $t$. Then, *regret* is defined:

$$L(\pi, \omega) = G(\pi_\omega^\star, \omega) - G(\pi, \omega) , \qquad (1)$$

where $\pi_\omega^\star$ is the optimal reward-maximizing policy under $\omega$. In our robust setting, our objective is to compute a policy $\pi^\dagger$ that minimizes the maximum regret $L$ possible for any realization of $\omega$, i.e.:

$$\pi^\dagger = \min_\pi \max_\omega L(\pi, \omega) . \qquad (2)$$

This problem is computationally expensive to solve since simply computing a policy $\pi$ that maximizes the reward $G(\pi, \omega)$ is PSPACE-hard [Papadimitriou and Tsitsiklis, 1994] even when the $T_n$ are known, i.e., $\omega$ is given.

A more tractable approach for computing multi-action RMAB policies $\pi$ is to utilize the Lagrangian relaxation [Hawkins, 2003, Killian et al., 2021b], reproduced below. For a given $\omega$, the optimal policy $\pi_\omega^\star$ maximizes the constrained Bellman equation:

$$J(\boldsymbol{s}) = \max_{\boldsymbol{A}^c} \left\{ \sum_{n=1}^{N} R(\boldsymbol{s}_n) + \beta \underset{\omega}{\mathbb{E}}[J(\boldsymbol{s}') \mid \boldsymbol{s}, \boldsymbol{A}^c] \right\} \qquad (3)$$

where $\boldsymbol{A}^c \subseteq \boldsymbol{A}$

$$\text{s.t.} \sum_{n=1}^{N} \sum_{j=1}^{|\mathcal{A}|} \boldsymbol{A}_{nj} c_j \leq B \qquad \sum_{j=1}^{|\mathcal{A}|} \boldsymbol{A}_{nj} = 1 \ \forall n \in [N]$$

where $\boldsymbol{A}_{nj} = 1$ if the $j^{\text{th}}$ action is taken on arm $n$ (else 0) and $c_j \in \mathcal{C}$ is the $j^{\text{th}}$ action cost. We then take the Lagrangian relaxation of the budget constraint [Hawkins, 2003], giving:

$$J(\boldsymbol{s}, \lambda^\star) = \min_\lambda \left( \frac{\lambda B}{1 - \beta} + \sum_{n=1}^{N} \max_{j \in |\mathcal{A}|} \{Q_n(\boldsymbol{s}_n, a_{nj}, \lambda)\} \right) \qquad (4)$$

$$\text{where } Q_n(\boldsymbol{s}_n, a_{nj}, \lambda) = R(\boldsymbol{s}_n) - \lambda c_j + \\ \beta \mathbb{E}_\omega \left[ Q_n(\boldsymbol{s}_n', a_{nj}, \lambda) \mid \pi_\omega^{La}(\lambda) \right] . \qquad (5)$$

Here, $a_{nj}$ is the $j^{\text{th}}$ action of arm $n$, $Q$ is the state-action value function, and $\pi_\omega^{La}(\lambda)$ is the optimal policy for a given $\lambda$. The key insight is that this relaxation decouples the value functions of the arms, except for the shared $\lambda$, i.e., for a given value of $\lambda$, all $Q_n$ could be solved via $N$ individual value iterations. However, finding and setting $\lambda := \lambda^\star$ is critical to finding good policies for multi-action RMABs [Killian et al., 2021b, Glazebrook et al., 2011], where $\pi_\omega^{La}(\lambda^\star)$

---

[1]Our approaches also easily extend to weakly-coupled MDPs, which allow multiple budget constraints [Hawkins, 2003], as well as to continuous-action RMABs, previously unstudied.

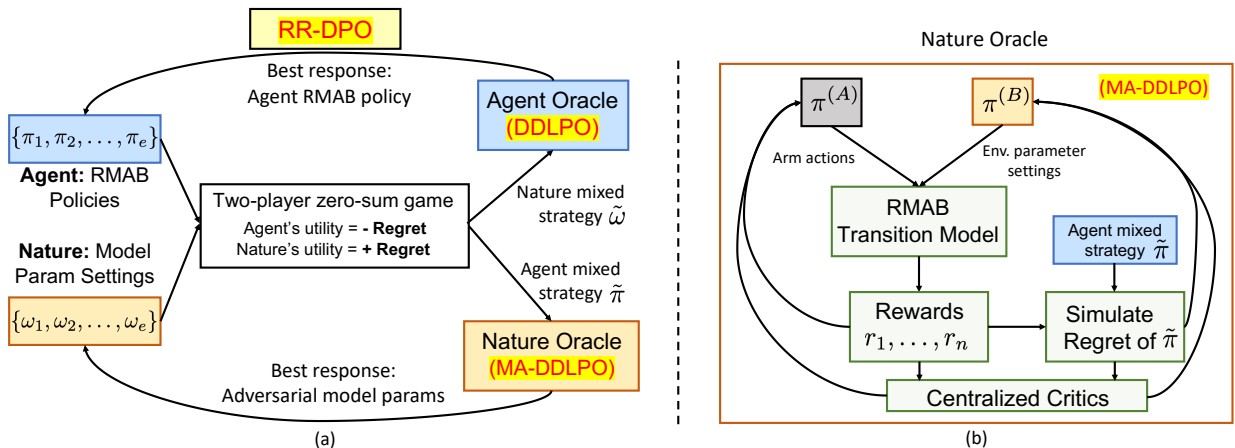

Figure 1: **(a)** Proposed framework for solving the Robust RMAB problem. The main loop follows a DO approach to iteratively compute a minimax regret optimal RMAB policy where each oracle is a novel DRL algorithm for RMABs. **(b)** The nature oracle: a novel multi-agent RL formulation of RMAB, that tackles non-stationarity with a centralized critic.

is used to recover a policy that respects the original budget constraint by solving a knapsack with $Q_n(\boldsymbol{s}_n, a_{nj}, \lambda^\star)$ as values, $\mathcal{C}$ as weights, and the constraints of Eq. 3, then taking the actions according to the $Q_n$ in the solved knapsack. The knapsack solution finds the combination of actions with the largest sum of learned $Q_n(\boldsymbol{s}_n, a_{nj}, \lambda^\star)$ values which still respects the budget. The integer program for the knapsack is given in Appendix B and has time complexity $\mathcal{O}(N|\mathcal{A}|B)$ [Killian et al., 2021b].

## 4 SOLVING ROBUST RMABs

We now build our approach for finding robust RMAB policies, visualized in Fig. 1(a). We use an iterative DO approach which achieves the minimax regret objective of Eq. 2 by casting the optimization problem as a zero-sum game between two players: an *agent* which learns policies $\pi$ to minimize regret, and an adversarial *nature* which selects environment parameters $\omega$ to maximize regret of the agent. In this two-player game, the *pure strategy* space for the agent is the set of all feasible RMAB policies $\pi$ that respect the budget constraint. The pure strategy space for nature is a continuous, closed set of parameters $\omega$ within the given uncertainty intervals. The algorithm maintains a pure strategy set for the agent and nature (Fig. 1(a) left boxes); each iteration, these strategy sets are used to compute a *mixed strategy*—i.e., a probability distribution over pure strategies—Nash equilibrium in a regret game (Fig. 1(a) center). Each oracle then learns a best response against the opponent's mixed strategy to add to its strategy set (Fig. 1(a) right boxes).

The agent oracle's goal is to find an RMAB policy $\pi$, or pure strategy, to minimize regret (Eq. 1) given a nature mixed strategy $\tilde{\omega}$. That is, the agent minimizes $L(\pi, \tilde{\omega})$ w.r.t. $\pi$, while $\tilde{\omega}$ is constant. Recall from Eq. 1 that $L(\pi, \tilde{\omega}) =$

$G(\pi_{\tilde{\omega}}^\star, \tilde{\omega}) - G(\pi, \tilde{\omega})$. Since $\tilde{\omega}$ and $\pi_{\tilde{\omega}}^\star$ are constant, then the first term $G(\pi_{\tilde{\omega}}^\star, \tilde{\omega})$ is also constant. Thus minimizing $L(\pi, \tilde{\omega})$ is equivalent to maximizing the second term $G(\pi, \tilde{\omega})$, which is maximal at $\pi = \pi_{\tilde{\omega}}^\star$. In other words, the agent oracle must compute an optimal reward-maximizing policy w.r.t. $\tilde{\omega}$. Such a reward-maximizing objective aligns with existing RL techniques, but still requires that we address the challenge of learning in the combinatorial state and action spaces of the RMAB. To address this challenge, *we propose a new RL method which decomposes the RMAB into $N$ per-arm learning problems and a complementary $\lambda$-network learning problem*, which together learn to spend limited budget where it will give the best return, detailed in Sec. 4.1.

Conversely, the nature oracle seeks to find a parameter setting $\omega$, or pure strategy, that maximizes the agent's regret given a mixed strategy $\tilde{\pi}$, i.e., maximize $L(\tilde{\pi}, \omega)$ with respect to $\omega$, while $\tilde{\pi}$ is fixed. This objective is even more challenging because both $G(\pi_\omega^\star, \omega)$ and $G(\tilde{\pi}, \omega)$ are functions of $\omega$. Most critically, computing $G(\pi_\omega^\star, \omega)$ requires obtaining an optimal policy $\pi_\omega^\star$ as $\omega$ changes in the optimization—this amounts to a planning problem in which an agent must learn an optimal policy while the environment changes, controlled by $\omega$, making the nature oracle difficult to solve. Moreover, in the interval uncertainty setting we consider, $\omega$ is defined by a space of continuous values; thus nature's pure strategy space is infinite, making the problem even more complex, since it cannot be exhaustively searched.

*To tackle this complexity we propose a novel method for implementing the regret-maximizing nature oracle by casting it as an MARL problem.* The approach, visualized in Fig. 1(b), trains one auxiliary agent to solve for a policy $\pi_\omega^\star$ ($\pi^A$ in Fig. 1(b)), needed to compute $G(\pi_\omega^\star, \omega)$ in the regret term, and simultaneously trains a second agent to learn worst-case parameters $\omega$ ($\pi^B$ in Fig. 1(b)) that minimize

**Algorithm 1** DDLPO

**Input**: Initial state $\boldsymbol{s}_0$, nature mixed strategy $\tilde{\omega}$, `n_epochs, n_subepochs, n_steps`

1: Init. policy networks $\theta_n$ for each arm $n \in [N]$
2: Init. critic networks $\phi_n$ for each arm $n \in [N]$
3: Init. $\lambda$-network $\Lambda$
4: Init. `buff` = [] and $\boldsymbol{s} = \boldsymbol{s}_0$
5: **for** $epoch = 1, 2, \ldots,$ `n_epochs` **do**
6:     Sample $\lambda = \Lambda(\boldsymbol{s})$
7:     Sample $\omega \sim \tilde{\omega}$
8:     **for** $subepoch = 1, \ldots,$ `n_subepochs` **do**
9:         **for** timestep $t = 1, \ldots,$ `n_steps` **do**
10:           Sample actions $a_n \sim \theta_n(s_n, \lambda)\ \ \forall n \in [N]$
11:           $\boldsymbol{s}', \boldsymbol{r} = $ `Simulate`$(\boldsymbol{s}, \boldsymbol{a}, \omega)$
12:           Add tuple $(\boldsymbol{s}, \boldsymbol{a}, \boldsymbol{r}, \boldsymbol{s}', \lambda)$ to `buff`
13:           $\boldsymbol{s} = \boldsymbol{s}'$
14:         Update each $(\theta_n, \phi_n)$ pair via PPO, using trajectories in `buff`
15:     Update $\Lambda$ via Prop. 1 with costs of final subepoch
16: **return** $\theta_1, \ldots, \theta_N, \phi_1, \ldots, \phi_N$ and $\Lambda$

---

**Algorithm 2** DDLPO-Act

**Input**: State $\boldsymbol{s}$, costs $\mathcal{C}$, budget $B$, agent actor, critic, and $\lambda$ networks $\theta_1, \ldots, \theta_N, \phi_1, \ldots, \phi_N, \Lambda$, selection method $\alpha$

1: $\lambda = \Lambda(\boldsymbol{s})$
2: **if** $\alpha ==$ 'GreedyProba' **then**
3:     $p_n = \theta_n(s_n, \lambda)\ \ \forall n \in [N]$ *// Action distr. of arm $n$*
4:     $\boldsymbol{a} = $ `GreedyProba`$(\boldsymbol{p}, \mathcal{C}, B)$     *// Greedily select highest probability actions until budget $B$ is reached*
5: **else if** $\alpha ==$ 'QKnapsack' **then**
6:     $q_{nj} = \phi_n(s_n, a_{nj}, \lambda)\ \ \forall n \in [N], \forall j \in [|\mathcal{A}|]$
7:     $\boldsymbol{a} = $ `QKnapsack`$(\boldsymbol{q}, \mathcal{C}, B)$     *// Solve knapsack in Appendix B*
8: **else if** $\alpha ==$ 'Whittle' **then**     *// Binary action only*
9:     $\boldsymbol{a} = $ BINASEARCH$(\boldsymbol{s}, B, \phi_1, ..., \phi_N)$ *// Appendix B*
10: **return** $\boldsymbol{a}$

---

$G(\tilde{\pi}, \omega)$—together, these will maximize the regret $L(\tilde{\pi}, \omega)$. With this MARL setup, we mitigate nonstationarity through centralized critic networks which allow each agent to include the other's actions in their learned state space. Solving a MARL problem requires an RL algorithm to optimize the underlying policy, so we first introduce our novel RL approach, DDLPO, to solve RMABs (Sec. 4.1) as a part of our agent oracle and then use the algorithm as the backbone of our nature oracle (Sec. 4.2).

### 4.1 AGENT ORACLE: DEEP RL FOR RMAB

Existing DRL approaches can be applied to the objective in Eq. 3 but, as detailed in Sec. 2, they fail to scale past trivially sized RMAB problems since the action and state spaces grow exponentially in $N$. To overcome this, we develop a novel DRL algorithm that instead solves the decoupled problem (Eq. 4). The key benefit of decoupling is to render policies and $Q$ values of each arm independent, allowing us to learn $N$ independent networks with *linearly sized state and action spaces, relieving the combinatorial burden of the learning problem*. However, this decoupling approach introduces a new technical challenge in solving the dual objective which maximizes over policies but minimizes over $\lambda$, as discussed in Sec. 3.

To solve this, we derive a dual gradient update procedure that iteratively optimizes each objective as follows: (1) holding $\lambda$ constant, learn $N$ independent policy networks via policy gradient, augmenting the state space to include $\lambda$ as input, as in Eq. 4; (2) use sampled trajectories from those learned policies as an estimate to update $\lambda$ towards its min-

imizing value via a novel gradient update rule. Another challenge is that $\lambda^\star$ of Eq. 4 depends on the current state of each arm—therefore, a key element of our approach is to learn this function $\lambda^\star(\boldsymbol{s})$ concurrently with our iterative optimization, using a neural network we call the $\lambda$-network that is parameterized by $\Lambda$. To train the $\lambda$-network, we use the following gradient update rule.

**Proposition 1.** *To learn the value $\lambda$ that minimizes Eq. 4 given a state $\boldsymbol{s}$, the $\lambda$-network, parameterized by $\Lambda$, should be updated with the following gradient rule:*

$$\Lambda_t = \Lambda_{t-1} - \alpha \left( \frac{B}{1-\beta} + \sum_{n=1}^{N} D_n(s_n, \lambda_{t-1}(\boldsymbol{s})) \right) \quad (6)$$

*where $\alpha$ is the learning rate and $D_n(s_n, \lambda)$ is the negative of the expected $\beta$-discounted sum of action costs for arm $n$ starting at state $s_n$ under the optimal policy for arm $n$ for a given value of $\lambda$.*

As $D_n$ lacks a closed form, the key insight we make is that it can be estimated by sampling multiple rollouts of the policy networks of all arms during training. As long as arm policies are trained for adequate time on the given value of $\lambda$, the gradient estimate will be accurate, i.e., $D_n(s_n, \lambda_{t-1}(\boldsymbol{s})) \approx - \sum_{k=0}^{K-1} \beta^k c_n^k$ where $K$ is the number of samples collected in an epoch and $c_n^k$ is the action cost of arm $n$ in round $k$. Moreover, this procedure will converge to the optimal parameters $\Lambda^\star$ if the arm policies are optimal.

**Proposition 2.** *Given arm policies corresponding to optimal $Q$-functions, Prop. 1 will lead $\Lambda$ to converge to the optimal as the number of training epochs and $K \to \infty$.*

Proofs are given in Appendix A. One interesting feature of this update rule is that to collect samples that reflect the proper gradient, the RMAB budget must not be imposed *at training time*—rather, the policy networks and $\lambda$-network

must be allowed to learn to play the Lagrange policy of Eq. 4, which learns to spend the correct budget in expectation, via our iterative update procedure. Therefore, at training time, we sample actions randomly according to the actor network distributions, without imposing the budget constraint. However, *at test time, we always take actions in a way that respects the budget constraint* as described in Alg. 2. Alg. 2 chooses actions either by (1) selecting greedily by the probabilities of the arm actor networks (2) using the learned $Q(\lambda)$-functions of the arm critic networks to follow the Q-value-maximizing knapsack procedure (Appendix B), or (3) in binary-action settings, using the $Q(\lambda)$-functions to follow a binary search procedure such that selected actions are equivalent to the Whittle index policy (Appendix B).

In theory, the policy networks could be trained via any DRL procedure that ensures the above characteristics for training the $\lambda$-network. In practice, we train with proximal policy optimization (PPO) [Schulman et al., 2017], a state-of-the-art policy gradient approach. Importantly, PPO is also flexible enough to handle both discrete and continuous actions which is necessary for the nature oracle.

Finally, to enable our iterative, dual-update procedure in practice, we need a mechanism to both (1) explore new arm policy actions after an update to $\Lambda$, then (2) exploit learned policy actions to develop good gradient estimates for $\Lambda$. We navigate this important trade-off by adding an entropy regularization term to the policy networks losses, controlled via a cyclical temperature parameter. We call our algorithm Deep Distributed Lagrange Policy Optimization (DDLPO), provide pseudocode in Algorithm 1, and include more implementation details in Appendix D.

## 4.2   NATURE ORACLE: MULTI-AGENT RL

Armed with a DRL procedure for learning RMAB policies, we now develop the MARL procedure, which we call MA-DDLPO, to implement the nature oracle. Recall that the challenge of the nature oracle is to jointly optimize a policy $\pi_\omega^\star$ and environment parameters $\omega$. We propose to solve this optimization using MARL, designed to handle this form of non-stationarity [Lowe et al., 2017] via centralized critics. In our MARL setup, each of two "players" (i.e., the "multiple agents") will aim to compute $\pi_\omega^\star$ and $\omega$, respectively, with separate objectives. The procedure is visualized in Fig. 1(b).

To implement the MARL nature oracle, we introduce two new players $A$ and $B$. Player $A$ is an *auxiliary player* whose goal is to optimize the RMAB policy $\pi_\omega^\star$ given a changing $\omega$, i.e., the first term of regret (Eq. 1. We call $A$ auxiliary because its learned policy will never be used outside the nature oracle; $A$ is only used to assist the nature oracle in computing the regret associated with a given $\omega$. Alternatively player $B$ is an adversarial player whose goal is the same as that of the nature oracle itself, i.e., to find parameters

---

**Algorithm 3** MA-DDLPO
___
**Input**: Agent mixed strategy $\tilde{\pi}$, `n_epochs`, `n_subepochs`, `n_steps`, `n_sims`
___
1: Init. player A: arm policy networks $\theta_n^{(A)}$ and arm critic networks $\phi_n^{(A)} \; \forall n \in [N]$, and $\lambda$-network $\Lambda$
2: Init. player B: environment parameter policy network $\theta^{(B)}$, critic network $\phi^{(B)}$
3: Init. `buff = []`
4: **for** *epoch* $= 1, 2, \ldots,$ `n_epochs` **do**
5:     Sample $s$ at random
6:     Sample $\lambda = \Lambda(s)$
7:     **for** *subepoch* $= 1, \ldots,$ `n_subepochs` **do**
8:         **for** $t = 1, \ldots,$ `n_steps` **do**
9:             Sample $a_n^{(A)} \sim \theta_n^{(A)}(s_n, \lambda)$ for each $n \in [N]$
10:            Sample $\omega^{(B)} \sim \theta^{(B)}(s)$
11:            $r^{(A)}, s' = \text{SIMULATE}(s, a^{(A)}, \omega^{(B)})$
12:            $\tilde{r} = \text{SIMULATE}(s, \tilde{\pi}(s), \omega^{(B)}, \text{n\_sims})$
                   *// (mean of n_sims 1-step rollouts of $\tilde{\pi}$)*
13:            $r^{(B)} = \left( \sum_{n \in [N]} r_n^{(A)} \right) - \tilde{r}$     *// (regret of $\tilde{\pi}$)*
14:            Add $(s, a^{(A)}, \omega^{(B)}, r^{(A)}, r^{(B)}, s', \lambda)$ to `buff`
15:            $s = s'$
16:        Update each $(\theta_n^{(A)}, \phi_n^{(A)})$ pair using trajectories in `buff`. $\phi_n^{(A)}$ get $\omega^{(B)}$ as part of state
17:        Update $\Lambda$ via Prop. 1 with costs of final subepoch
18:        Update $\theta^{(B)}, \phi^{(B)}$ using trajectories in `buff`. $\phi^{(B)}$ gets $a^{(A)}$ as part of state
19: **return** $\theta^{(B)}$
___

$\omega$ that maximize regret of the current agent mixed strategy $\tilde{\pi}$. We define a shared transition function for the environment in which the players act $T : \mathcal{S} \times \mathcal{A}_A \times \mathcal{A}_B \to \mathcal{S}$. Here, $\mathcal{A}_A$ is the action space of the underlying multi-action RMAB. At a given state $s$, the action space $\mathcal{A}_B$ defines for player $B$ actions $\omega$ which, in general, depend on $s$. That is, at each step, player $B$ selects environment parameters $\omega$, and thus transition probabilities that will influence the outcome of player $A$'s actions. We adopt the centralized critic idea from multi-agent PPO [Yu et al., 2021] to our RMAB setting to create MA-DDLPO. A notable strength of our MARL approach is that it allows the discrete-space policy of player $A$ and the continuous-space policy of player $B$ to be learned by separate networks, simplifying training compared to an alternative combined-network approach. Moreover, our choice to use PPO offers a convenient way to learn both types of policies as separate networks, while utilizing a single framework of update rules.

A critical step is then to define the rewards for players $A$ and $B$ to match their objectives. Since player $A$'s objective is to find $\pi_\omega^\star$, it adopts the reward defined by the underlying RMAB, i.e., $R^{(A)}(s) = \sum_{n=1}^N R_n(s)$. However, player $B$'s objective is to learn the regret-maximizing parameters $\omega$. This objective is challenging because it requires computing

---
**Algorithm 4** RR-DPO
---
**Input**: Environment simulator and parameter uncertainty intervals $\overline{\omega}_n$ for all $n \in [N]$
**Parameters**: Convergence threshold $\varepsilon$
**Output**: Agent mixed strategy $\tilde{\pi}$

1: $\Omega_0 = \{\omega_0\}$, with $\omega_0$ selected at random
2: $\Pi_0 = \{\pi_{B_1}, \pi_{B_2}, \ldots\}$, where $\pi_{B_i}$ are baseline and heuristic strategies
3: **for** epoch $e = 1, 2, \ldots$ **do**
4:     Solve for $(\tilde{\pi}_e, \tilde{\omega}_e)$, mixed Nash equilibrium of regret game with strategy sets $\Omega_{e-1}$ and $\Pi_{e-1}$
5:     $\pi_e = \text{DDLPO}(\tilde{\omega}_e)$
6:     $\omega_e = \text{MA-DDLPO}(\tilde{\pi}_e)$
7:     $\Omega_e = \Omega_{e-1} \cup \{\omega_e\}, \Pi_e = \Pi_{e-1} \cup \{\pi_e\}$
8:     **if** $L(\tilde{\pi}_e, \omega_e) - L(\tilde{\pi}_{e-1}, \tilde{\omega}_{e-1}) \leq \varepsilon$ and $L(\pi_e, \tilde{\omega}_e) - L(\tilde{\pi}_{e-1}, \tilde{\omega}_{e-1}) \leq \varepsilon$ **then**
9:         **break**
10: **return** $\tilde{\pi}_e$
---

and optimizing over the returns of the fixed input policy $\tilde{\pi}$ with respect to all possible $\omega$, which is in general non-convex. In practice, to estimate the returns of $\tilde{\pi}_\omega$, we execute a series of roll-outs against player $B$'s current action. That is, given $s$ at a given round, we sample an action from $\tilde{\pi}_\omega$ and the next state $s'$, and define the *regret-based* reward of player $B$, as $R^{(B)} = \sum_{n=1}^N R_n(s_n) - \frac{1}{Y} \sum_{y=1}^Y r_y^{\tilde{\pi}, \omega}$, where $r_y^{\tilde{\pi}, \omega}$ is the reward from each of $Y$ one-step Monte Carlo simulations of the mixed strategy $\tilde{\pi}$ in $\omega$.

To train the policies, player $A$ has the same policy network architecture as DDLPO, i.e., $N$ discrete policy networks and one $\lambda$-network, and the player $B$ actor network is a single continuous-action policy network. Since players $A$ and $B$ have separate reward functions, they have their own critic networks, but these critics are *centralized* in that they both take the actions of the other as input. Other than the centralized critic, player $A$ is trained the same way as DDLPO, and player $B$ is trained in a standard PPO fashion. In practice, to ensure good gradient estimates for player $A$'s $\lambda$-network in MA-DDLPO, we keep player $B$'s network—and thus the environment—constant between $\Lambda$ updates, updating $B$'s network with the same frequency as the $\lambda$-network updates. Pseudocode for MA-DDLPO is given in Alg. 3 and further details of its implementation are given in Appendix D.

### 4.3 MINIMAX REGRET RMAB DOUBLE ORACLE

We now have all the pieces needed to run our robust algorithm, Robust RMABs via Deep Policy Oracles (RR-DPO), visualized in Fig. 1(a), with pseudocode presented in Algorithm 4, adapted from the MIRROR framework [Xu et al., 2021]. We use DDLPO to instantiate the agent oracle, MA-DDLPO for the nature oracle, and run RR-DPO until the improvement in value for each player is within a tolerance $\varepsilon$

or until a set number of iterations.

We now establish conditions under which RR-DPO converges to the minimax regret–optimal policy in finite iterations. In the binary-action setting, assuming each oracle returns true best responses, and under an analytical condition that is straightforward to achieve, i.e., finite pure strategy sets:[2]

**Proposition 3.** *RR-DPO converges in a finite number of steps to the minimax regret-optimal policy.*

In addition, we empirically verify that good policies are found outside of these conditions, and that RR-DPO converges using our continuous-strategy-space nature oracle. Further, we show that a policy that maximizes reward assuming a fixed parameter set can incur arbitrarily large regret when the parameters are changed (proofs in Appendix A).

**Proposition 4.** *In the Robust RMAB problem with interval uncertainty, the max regret of a reward-maximizing policy can be arbitrarily large compared to a minimax regret-optimal policy.*

## 5 EXPERIMENTAL EVALUATION

We first experimentally demonstrate the importance of robust planning in the presence of uncertainty using a hand-crafted synthetic domain (inspired by Prop. 4). We then evaluate our algorithm on two challenging real-world-inspired public health planning scenarios which demonstrate the capability of our robust RMAB framework. All experiments use selection method $\alpha =$ 'GreedyProba' for DDLPO-Act (Alg. 2), which we found had the best performance.

We compare RR-DPO against five baselines. These baselines include three variations of the reward-maximizing approach from Hawkins [2003], which, given fixed environment parameters $\omega$, at each step computes a Lagrange policy, then chooses actions following the knapsack procedure described in Sec. 3. The three variations are pessimistic (**HP**), mean (**HM**), and optimistic (**HO**), which assume the environment parameters are set at the lower bound, mean, and upper bound of the given intervals for each arm. We also implement **RLvMid**, which *learns* (rather than computes) a policy via DDLPO assuming *mean* parameters, and **Rand**, which acts randomly to fill the budget. All results are averaged over 50 random seeds and were executed on a cluster running CentOS with Intel(R) Xeon(R) CPU E5-2683 v4 @ 2.1 GHz with 8GB of RAM using Python 3.7.10. Our DDLPO implementation builds on OpenAI Spinning Up [Achiam, 2018] and RR-DPO builds on the MIRROR implementation [Xu et al., 2021], computing Nash equilibria using Nashpy 0.0.21 [Knight and Campbell, 2018]. Code is available at https://github.com/killian-34/RobustRMAB and hyperparameter settings are in Appendix D.

---
[2]Straightforward to achieve for nature oracle via discretization.

## 5.1 EXPERIMENTAL DOMAINS

**Synthetic** demonstrates that reward-maximizing policies (RLvMid, HP, HM, HO) may incur large regret in the presence of uncertainty. There are three binary-action arm types $\{U, V, W\}$, each with $\mathcal{C} = \{0, 1\}$, $\mathcal{S} = \{0, 1\}$, $R(s) = s$, and the following transition matrix, with rows and columns corresponding to actions and next states, respectively:

$$T^n_{s=0} = \begin{bmatrix} 0.5 & 0.5 \\ 0.5 & 0.5 \end{bmatrix}, \quad T^n_{s=1} = \begin{bmatrix} 1.0 & 0.0 \\ 1-p_n & p_n \end{bmatrix}$$

$$p_U \in [0.00, 1.00], \; p_V \in [0.05, 0.90], \; p_W \in [0.10, 0.95]$$

When an arm is at $s = 0$, each action has equal impact on the state transition. When the arms are at $s = 1$, selecting arms with high $p_n$ is optimal. This implies that policies can be specified by the order in which arms would be acted on, when they are in state $s = 1$. Accordingly, $\pi_{HP} = [W, V, U]$, $\pi_{HM} = [W, U, V]$, and $\pi_{HO} = [U, W, V]$. However, observe that there exist values of $p_n$ that can make each of the reward-maximizing policies incur large regret, e.g., for $\pi_{HO}$ $p_U = 0.0, p_V = 0.9, p_W = 0.1$ would induce an optimal policy $[V, W, U]$, which is the reverse of $\pi_{HO}$.

**ARMMAN** is a real-world *maternal healthcare intervention problem* modeled as a binary-action RMAB [Biswas et al., 2021]. The goal is to select a subset of mothers each week to intervene on to encourage engagement with automated maternal health messaging. The behavior of enrolled women is modeled by an MDP with three states: Self-motivated, Persuadable, and Lost Cause. We use the summary statistics given in their paper and assume uncertainty intervals of $0.5$ centered around the transition parameters, resulting in 6 uncertain parameters per arm (details in Appendix C.1). Similar to the setup by Biswas et al. [2021], we assume 1:1:3 split of arms with high, medium, and low probability of increasing their engagement upon intervention. In our experiments, we scale the value of $N$ in multiples of 5 to keep the same split of arm categories of 1:1:3.

**SIS Epidemic Model** is a discrete-state model in which arms represent distinct geographic regions and each member of an arm's population of size $N_p$ is either (**S**)usceptible to or (**I**)nfected with an infectious disease. Such models have been the subject of increased interest following the COVID-19 pandemic [Hinch et al., 2021, Kerr et al., 2021], and will represent a large-state and multi-action experimental domain. In our model, the count of **S** members of the population is the state of each arm. Each arm's SIS model is defined by parameters $\kappa$, the average number of contacts per round, and $r_{infect}$, the probability of infection given contact with an **I** member. Details on computing discrete state transition probabilities from these parameters are derived from Yaesoubi and Cohen [2011] and given in Appendix C.2. We introduce three intervention actions $\{a_0, a_1, a_2\}$ with costs $c = \{0, 1, 2\}$. Action $a_0$ represents no action, $a_1$ represents messaging about physical distancing (divides $\kappa$ by $a_1^{eff}$), and

$a_2$ represents distributing face masks (divides $r_{infect}$ by $a_2^{eff}$). We impose the following uncertainty intervals: $\kappa \in [1, 10]$, $r_{infect} \in [0.5, 0.99]$, $a_{\{1,2\}}^{eff} \in [1, 10]$.

## 5.2 PERFORMANCE OF RR-DPO

First, we evaluate the performance of the algorithms in uncertain environments. We compute the regret of an agent's pure strategy $\pi$ against a nature pure strategy $\omega$ as the difference in the average reward obtained by $\pi$ against $\omega$ and the average reward of the best strategy in the experiment against $\omega$. The average reward is the discounted sum of rewards over all arms for a horizon of length 10, over 25 simulations. In each setting, DO runs for 6 epochs, using 100 rollout steps and 100 training epochs for each oracle. After completion, each baseline strategy is evaluated by querying the nature oracle for the best response against that strategy, then computing max regret against all $\omega$. The regret of RR-DPO is computed as the utility of the agent mixed strategy returned by the DO over the two-player regret game.

Fig. 2(a–f) shows RR-DPO incurs the lowest regret, beating the baselines in all domains. (a,b) shows results on the synthetic domain, demonstrating our approach can reduce regret by ~50% against the benchmarks, across various values of $N$ and $B$. Moreover, as $B$ increases, the regret incurred may increase, since higher budget implies better reward potential for the optimal policy; however, the regret for RR-DPO remains small even as $B$ grows. Similarly, for the ARMMAN domain (c,d), a challenging domain adapted from a real-world problem, our algorithm performs consistently better than the baselines, achieving regret that is around 50% lower than the best baselines. In the SIS domain (e–f), another real-world planning setting with a larger state space and multiple actions, our results are robust across parameter settings. Importantly, this holds even as we increase the state space from $S = 100$ to $500$ (f), in which running the Hawkins baseline becomes prohibitively expensive.

*Finally, we run sensitivity analyses of the algorithms against H and the size of the uncertainty sets* (Appendix Fig. A1). When $H$ varies from 10 to 100, RR-DPO maintains very low regret, while competitor regret as much as doubles, increasing RR-DPO's relative improvement as high as ~60%. Similar results are obtained when varying the uncertainty intervals between 0.25, 0.5 and 1.0 times their widths from the experiments in Fig. 2, with RR-DPO always dominating.

## 5.3 PERFORMANCE OF DDLPO

We also evaluate the performance of DDLPO, our novel DRL approach to find reward-maximizing policies for multi-action RMABs, which implements our agent oracle. We compare against **No Action** and **Random** baselines as well as the computationally intensive solution by Hawkins which

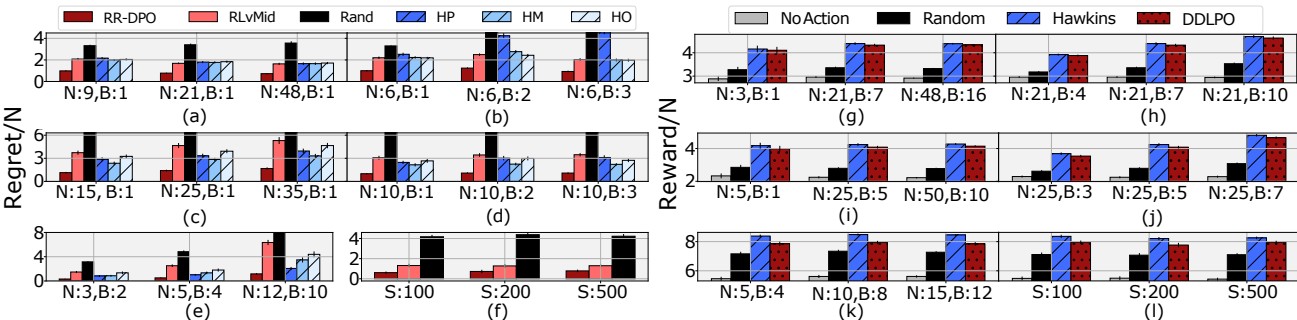

Figure 2: **(a–f)** Maximum policy regret of RR-DPO in robust setting for Synthetic (a,b), ARMMAN (c,d) and SIS (e,f) domains. Lower is better. Synthetic is scaled by 3 and ARMMAN by 5 to maintain the distributions of arm types specified in Sec. 5. (e) uses $S = 50$ and (f) uses $N = 5, B = 4$. RR-DPO beats all baselines by a large margin across various settings. **(g–l)** Returns of DDLPO for reward-maximizing setting (agent oracle) for synthetic (g,h), ARMMAN (i,j), and SIS (k,l) domains. Higher is better. (k) uses $S = 50$ and (l) uses $N = 5, B = 4$. DDLPO is competitive across parameter settings.

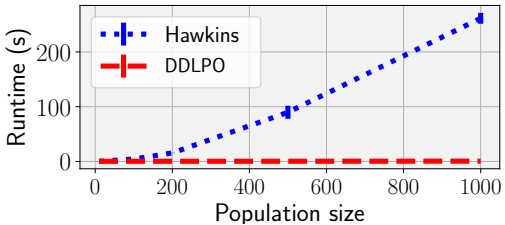

Figure 3: The poor scaling of query time of the Hawkins baseline compared to DDLPO, as discussed in Sec. 5, even for relatively small problem sizes ($N = 10, B = 2$).

computes the Lagrange policy, but which requires exact environment parameters and discrete states/actions. Hawkins upper bounds DDLPO for small discrete problems since it is exact whereas DDLPO learns the Lagrange policy from samples. Each experiment is a traditional reward-maximizing RMAB instantiated with a random sample of valid parameter settings for each seed.

Fig. 2(g–l) shows DDLPO achieves reward comparable to the Hawkins algorithm and significantly better than random, providing insight into the success of our RR-DPO approach which DDLPO enables, and showing promise for DDLPO as an algorithm of general interest. In the synthetic domain (g,h), DDLPO learns to act on the 33% of arms who belong to category $W$. The mean reward of DDLPO almost matches that of Hawkins algorithm as $N$ scales with a commensurate budget (g). As we fix $N$ and vary the budget (h), the optimal policy accumulates more reward, and DDLPO almost equals the optimal. We observe similar results on the ARMMAN domain (i,j), a challenging real-world health problem. On the SIS domain (k,l), the strong performance of DDLPO holds in a multi-action setting even as we increase the number of states from 50 to 500 (l).

Moreover, DDLPO beats Hawkins computationally: in Fig. 3, a single rollout (10 rounds) of Hawkins takes ~100 seconds when there are 500 states, scaling quadratically in

general. This demonstrates that it would be prohibitive to run Hawkins in the loop of RR-DPO, since agent policies are evaluated thousands of times to compute the regret matrices. For just 25 simulations, computation would take ~42 minutes to evaluate a single cell in the regret matrix, which has $|\Pi| \times |\Omega|$ total cells.

# 6  CONCLUSION

We address a key limitation blocking RMABs from many real-world settings: that arm dynamics are not known precisely. To plan safe, effective policies, robust approaches accounting for uncertainty are essential, which we give in RR-DPO, enabled by DDLPO, a novel deep-RL algorithm for RMABs of general interest. We hope our contributions bring us closer to deploying RMABs for real-world impact.

**Author Contributions**

J.A.K. conceived and implemented algorithmic ideas, wrote code, designed and ran experiments, wrote proofs, created figures, and led writing the paper. L.X. contributed algorithmic ideas, wrote code, wrote proofs, and contributed to writing the paper. A.B. contributed algorithmic ideas and contributed to writing the paper. M.T. contributed guidance on the direction of the paper, contributed algorithmic ideas, and contributed to writing the paper.

**Acknowledgements**

This work was supported in part by the Army Research Office by Multidisciplinary University Research Initiative (MURI) grant number W911NF1810208. J.A.K. was supported by an NSF Graduate Research Fellowship under grant DGE1745303. A.B. was supported by the Harvard Center for Research on Computation and Society. Thank you to Andrew Perrault for feedback on an earlier draft.

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
