# OpenReview forum: "Restless and Uncertain: Robust Policies for Restless Bandits via Deep Multi-Agent Reinforcement Learning"
_auai.org/UAI/2022/Conference — UAI 2022 Poster_

### Official Review · Reviewer_vBS2 · 2022-04-12

**Q2(1) Originality/Novelty:** 2
**Q2(2) Significance/Impact:** 3
**Q2(3) Correctness/Technical Quality:** 3
**Q2(6) Clarity Of Writing:** 3
**Q6 Overall Score:** 7
**Q8 Confidence In Your Score:** 3

**Q1 Summary And Contributions:**

This paper studies learning a robust policy in the framework of restless multi-armed bandit (RMAB).  The proposed approach uses a double oracle framework (oracles for agent and nature), which is computationally challenging due to the combinatorial nature of RMAB. The authors design a deep reinforcement learning (RL) algorithm which tackles the combinatorial challenge by learning a policy network per arm, which greatly reduces sample complexity.

**Q2 Assessment Of The Paper:**

More detailed information regarding each of these aspects is given below:

**Q2(4) Quality Of Experiments (Optional):**

3: Good: The experimental evaluation is adequate, and the results convincingly support the main claims.

**Q2(5) Reproducibility:**

3: Good: Key resources (e.g., proofs, code, data) are available and key details (e.g., proofs, experimental setup) are sufficiently well-described for competent researchers to confidently reproduce the main results.

**Q3 Main Strengths:**

Most of RMAB methods assume that the underlying system dynamics are known, which is rarely the case in many real-world settings. Transition dynamics of states are often estimated with significant uncertainty, and cannot be point-identified. This paper proposes methods to address this challenge by learning a robust policy that could achieve reasonable performance in the worst-case scenario. The proposed algorithm seems reasonable. The theoretical result in Prop 3 shows that the proposed method finds the minimax optimal policy in a finite number of iterations.

**Q4 Main Weakness:**

The presentation for Section 4.2 could be improved, which concerns with the algorithm that finds transition dynamics that maximize the regret. I understand there exist challenges in the space constraint. However, if permitted, it would be appreciated if the authors could include the detailed algorithm for MA-DDLPO in the main manuscript. Also, I notice that there is no convergence guarantee for MA-DDLPO. Could the authors further elaborate on this?

**Q5 Detailed Comments To The Authors:**

Please refer to Q4.

**Q7 Justification For Your Score:**

This paper proposes a novel algorithm for learning a robust policy in RMAB models. It address challenges when transition probabilities of RMAB are unknown, and bounded in a specific region. The proposed algorithm seems reasonable, and the authors provide a theoretical guarantee that it converges to the minimax optimal policy. While the presentation of this paper could be improved, its results should be useful across disciplines.

**Q9 Complying With Reviewing Instructions:**

1: Yes.

---

### Official Review · Reviewer_BGoo · 2022-04-14

**Q2(1) Originality/Novelty:** 3
**Q2(2) Significance/Impact:** 2
**Q2(3) Correctness/Technical Quality:** 3
**Q2(6) Clarity Of Writing:** 3
**Q6 Overall Score:** 6
**Q8 Confidence In Your Score:** 3

**Q1 Summary And Contributions:**

The paper considers the problem of multi-action RMABs under budge with interval transition probabilities. Authors proposes a double oracle approach, with agent and nature oracles, to cast the minimax objective as a zero-sum games. Empirical results show the promising accuracy and/or efficiency gains.

**Q2 Assessment Of The Paper:**

More detailed information regarding each of these aspects is given below:

**Q2(4) Quality Of Experiments (Optional):**

3: Good: The experimental evaluation is adequate, and the results convincingly support the main claims.

**Q2(5) Reproducibility:**

2: Fair: Key resources (e.g., proofs, code, data) are unavailable but key details (e.g., proof sketches, experimental setup) are sufficiently well-described for an expert to confidently reproduce the main results.

**Q3 Main Strengths:**

The paper proposes a novel algorithm for an interesting research question.

Convergence results are provided.

empirical evaluation is promising.





**Q4 Main Weakness:**

- it is not clear if a regret bound for the minimax regret optimal policy is bounded and its value.


**Q5 Detailed Comments To The Authors:**

- it would be better if authors can clarify oracles when they are first used. What assumptions and optimality do these oracles use and produce?
- "we always take actions in a way that respects the budget constraint by following the knapsack procedure described at the end of section 3" It is unclear to me if one has to solve a knapsack problem at test time? If so, what is the efficiency?
- are the datasets used publicly available?

**Q7 Justification For Your Score:**

novelty approach with strong empirical evaluation

**Q9 Complying With Reviewing Instructions:**

1: Yes.

---

### Official Review · Reviewer_YKs1 · 2022-04-14

**Q2(1) Originality/Novelty:** 2
**Q2(2) Significance/Impact:** 2
**Q2(3) Correctness/Technical Quality:** 3
**Q2(6) Clarity Of Writing:** 3
**Q6 Overall Score:** 6
**Q8 Confidence In Your Score:** 2

**Q1 Summary And Contributions:**

The paper introduces a generalization of restless multi-armed bandits which allows stochastic dynamics to be specified as uncertainty intervals. The proposed approach uses oracles for agent and nature. A deep RL algorithm, which implements the reward-maximizing agent oracle, is introduced. The regret-maximizing nature oracle is formulated as a multi-agent RL problem. The validity of this approach is supported by experimental results.

**Q2 Assessment Of The Paper:**

More detailed information regarding each of these aspects is given below:

**Q2(5) Reproducibility:**

3: Good: Key resources (e.g., proofs, code, data) are available and key details (e.g., proofs, experimental setup) are sufficiently well-described for competent researchers to confidently reproduce the main results.

**Q3 Main Strengths:**

1. The provided code is well-documented thus enhancing the reproducibility.

2. The presented approach seems suitable for the problem at hand and the experimental results are fair.



**Q4 Main Weakness:**

The explanation of their proposed approach could be more detailed, especially Section 4.1 and Section 4.2.

**Q5 Detailed Comments To The Authors:**

See answer to Q4.

**Q7 Justification For Your Score:**

This paper addresses a well-motivated problem and proposed solution seems to work well. However, I cannot vouch for the novelty and expected impact as this paper falls outside my domain of expertise,

**Q9 Complying With Reviewing Instructions:**

1: Yes.

---

### Decision · Program_Chairs · 2022-05-15

**Decision:**

Accept (Poster)

**Comment:**

Meta Review: This paper applies reinforcement learning (RL) to learning a robust policy in restless bandits, which performs well in the worst case. This is a major departure from the traditional approaches to (restless) bandits, where the policy is designed manually to have low regret in theory. While the authors analyzed an idealized variant of their approach, the strengths are generality and being data-adaptive. None of the reviewers had major concerns. I also looked at the paper. My comment is that the authors should present their work better in the context of other attempts to learn bandit policies from data, such as

Algorithms for the multi-armed bandit problem: https://arxiv.org/pdf/1402.6028.pdf

Differentiable meta-learning of bandit policies: https://proceedings.neurips.cc/paper/2020/file/171ae1bbb81475eb96287dd78565b38b-Paper.pdf

Policy gradient optimization of Thompson sampling policies: https://arxiv.org/pdf/2006.16507.pdf

I support acceptance of this paper.